# VBASS enables integration of single cell gene expression data in Bayesian association analysis of rare variants

Guojie Zhong[1,2], Yoolim A. Choi[3] & Yufeng Shen [1,3,4 ✉]

Rare or de novo variants have substantial contribution to human diseases, but the statistical power to identify risk genes by rare variants is generally low due to rarity of genotype data. Previous studies have shown that risk genes usually have high expression in relevant cell types, although for many conditions the identity of these cell types are largely unknown. Recent efforts in single cell atlas in human and model organisms produced large amount of gene expression data. Here we present VBASS, a Bayesian method that integrates single-cell expression and de novo variant (DNV) data to improve power of disease risk gene discovery. VBASS models disease risk prior as a function of expression profiles, approximated by deep neural networks. It learns the weights of neural networks and parameters of Gamma-Poisson likelihood models of DNV counts jointly from expression and genetics data. On simulated data, VBASS shows proper error rate control and better power than state-of-the-art methods. We applied VBASS to published datasets and identified more candidate risk genes with supports from literature or data from independent cohorts. VBASS can be generalized to integrate other types of functional genomics data in statistical genetics analysis.

[1] Department of Systems Biology, Columbia University Irving Medical Center, New York, NY, USA. [2] Integrated Program in Cellular, Molecular, and Biomedical Studies, Columbia University Irving Medical Center, New York, NY, USA. [3] Department of Biomedical Informatics, Columbia University Irving Medical Center, New York, NY, USA. [4] JP Sulzberger Columbia Genome Center, Columbia University Irving Medical Center, New York, NY, USA. ✉email: ys2411@cumc.columbia.edu

About 3% of children are born with congenital anomalies or will develop neurodevelopmental disorders (NDD)[1]. Given the severe consequence of these conditions on reproductive fitness, risk variants with large effect are under strong negative selection and therefore have low frequency in the population. Recent genetics studies identified hundreds of risk genes of these conditions, largely by rare de novo variants[2–11], however, the majority of risk genes remain unidentified[10,12–15], due to challenges in statistical power in analysis of rare variants[16]. This is because that de novo coding variants with large effect sizes, including gene disruption variants and damaging missense variants, usually have low mutation rates and very low allele frequency in a study[17]. Several recently published methods attempt to increase power using additional information. EncoreDNM[18], m-TADA[19] and M-DATA[20] are statistical models that improve power by leveraging pleiotropic effect across conditions. DECO[21] integrates pathways and gene sets information to prioritize risk genes. Here we describe a method that integrates gene expression data of normal tissues with genetic data to improve power of finding new risk genes through rare variants.

Cell-type specific gene expression has long been used qualitatively for interpretation of biological mechanisms in developmental biology and genetics. Previously we have shown that high expression in developing heart and diaphragm is associated with increased burden of de novo coding variants in congenital heart disease (CHD)[7] and congenital diaphragmatic hernia[9], respectively. We have also shown that cell-type specific expression in brain is associated with plausibility of autism spectrum disorders (ASD) risk genes[22,23]. It is clear that gene expression profile can inform association analysis of rare variants for risk gene discovery. However, the ability to improve power in gene discovery using expression data has been hindered by the lack of rigorous statistical methods and cell type specific expression data from relevant tissues during development. Recent efforts in cell atlas of human and model organisms have been generating large amount of single cell expression data of adult tissues[24,25] in addition to an increase in various developmental stages[26–29]. Here we describe a novel computational method that leverages expression data with probabilistic models to improve statistical power of risk gene discovery.

VBASS (Variational inference Bayesian ASSociation), takes a vector of expression profile, such as cell-type specific expression from single cell RNA-seq and models the priors of risk genes as a function of expression profile of multiple cell types. VBASS uses deep neural networks to approximate the function and uses semi-supervised variational inference to estimate the parameters. Although optimized for scRNA-seq data, VBASS could also be applied to bulk RNA seq data with a simplified framework. We compared the performances of VBASS with extTADA[30], a state-of-the-art Bayesian method which does not use expression data as input, under two conditions (bulk and scRNA-seq data) by both simulated and published de novo variants datasets to assess error control and statistical power. We also compared with DECO[21] on published de novo variants datasets to assess the difference between gene expression data and curated gene set data for power improvement.

## Results

### VBASS models disease risk association with both genetics and expression data.
VBASS is a Bayesian mixture model with learnable priors (Fig. 1). We model the number of genetic variants of interest (e.g., LGD, likely gene disruption, or Dmis, damage missense de novo variants) in the gene as a sample drawn independently through mixture of Poisson and Gamma-Poisson (Negative binomial) Distributions ("Methods"). Such Gamma-

Poisson distribution has been proved useful in modeling the sparse de novo variant data[12,30]. Instead of using a naïve prior that all genes share the same probability of being disease risk, we assume that this prior should be gene specific. And it could be inferred from the spatiotemporal expression data of fetal development of corresponding organ as disease related genes are likely to be involved with similar pathways and regulation processes[10]. In VBASS, we model this prior $\pi_g$ as a function of expression profiles that could be approximated with a neural network $f_E$ (Fig. 1, Supplementary Fig. 1, and "Methods"). With such approximation it is possible to take the advantage of the state-of-art stochastic gradient descent method[31]. Generally, VBASS will assign a higher disease risk to genes with relatively high expression in disease associated cell types while low expression in non-associated cell types, and vice versa.

VBASS could also take bulk RNA seq data of certain organ or cell type as input when prior knowledge of its disease risk association is available. For example, the increased burden of damage variants of high heart expression genes in CHD[7]. In that case $x_g$ is a scalar and $f_E$ could be parameterized by three parameters $(A, B, C)$ that corresponds for a linear transformation followed by sigmoid activation ("Methods"). This sigmoid-shape function could quantify the fact that genes with higher expression in the corresponding organ or cell type are more likely to harbor disease risk variants.

We trained VBASS in a semi-supervised manner with stochastic gradient descent method to estimate the parameters ("Methods"). While for the bulk version with scalar input, VBASS can be trained in a completely unsupervised manner with MCMC ("Methods"). The estimated parameters were used to calculate Posterior Probability of Association (PPA) and False Discovery Rate (FDR) for all genes ("Methods").

### VBASS showed better power than extTADA on simulated data with bulk RNA-seq expression.
We tested the performance of VBASS and extTADA on simulated CHD dataset ("Methods"). As expected, both models showed good false discovery control and local false discovery control (Supplementary Fig. 2). VBASS outperformed extTADA with better recall under same precision level (Fig. 2a) under sample sizes from 2645 to 20,000. Although the difference in power decreases with increasing sample size, VBASS still outperformed extTADA by roughly 10% increase of recall at sample size of 10,000, which is feasible for CHD in the next few years.

To test the power of VBASS with respect to the size of genes, we calculated the recall rate at same significance levels (FDR ≤0.05) on both models for genes with different mutation rates. VBASS showed better statistical power especially for genes with higher mutation rates under small sample sizes (Fig. 2b). As the sample size increases, the power difference of VBASS and extTADA becomes smaller on large genes, while VBASS still outperforms extTADA on medium-mutation-rate genes (Fig. 2b). Overall, our simulation results showed that VBASS can increase the statistical power for prioritizing disease risk genes by estimating risk prior as a function of expression.

### VBASS showed better power than extTADA on simulated data with scRNA-seq expression.
We ran VBASS and extTADA separately on the simulation dataset ("Methods"). Both models showed good false discovery control (Fig. 3a). Which means that the estimated error rates from both models are close to the real error rates. In other words, if out model identified 100 genes at significance level 0.05, the real error late will be close to 5%, which is 5 false positives. To test the statistical power of VBASS and extTADA, we plotted the precision-recall curve using the output

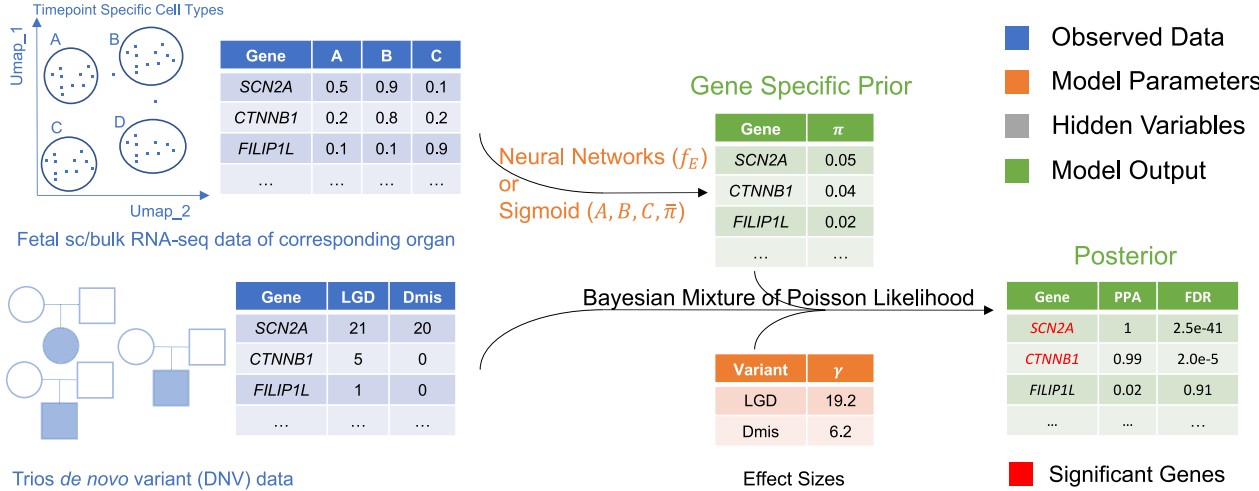

**Fig. 1 Model architecture of VBASS.** The input is a de novo variant table of patient cohort with either single cell or bulk expression profile of the corresponding organ during fetal development. VBASS will estimate the gene specific prior from the expression profile via neural networks (single cell version) or a sigmoid function (bulk version) and the effect sizes of different types of variants. It will then calculate the posterior probability of each gene's association with disease based on the estimated parameters. The significance is determined by False Discovery Rate (FDR) estimation for each gene based on the posterior probability.

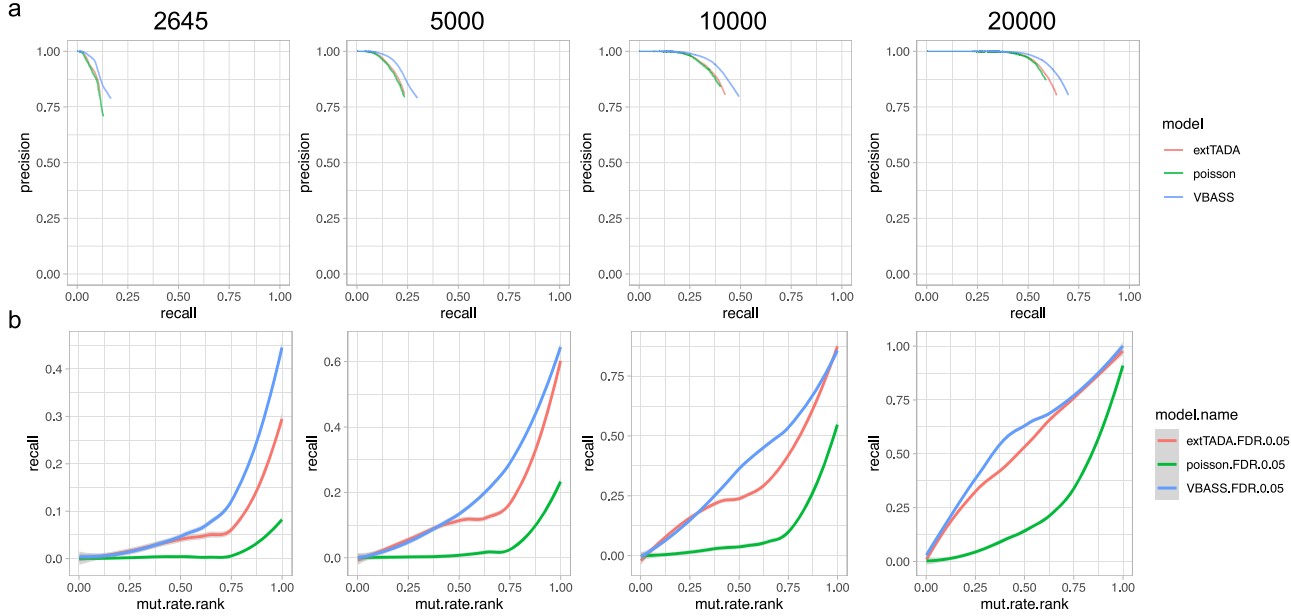

**Fig. 2 Performance comparison of bulk version VBASS and extTADA on simulation.** Figure titles are the cohort sizes during simulation. **a** Precision-recall in two models, only show the part with FDR ≤0.2 for extTADA and VBASS, only show the part with FDR ≤0.01 for Poisson test. **b** Comparison of recall (y-axis) for genes sets with different mutation rates (x-axis).

posterior probabilities from VBASS, extTADA and the real parameters we used in simulation. VBASS outperformed extTADA with higher recall under same precision (Fig. 3b). In other words, under same significance level (0.05 or 0.1), our model can identify more genes, while the error rate remains the same, as stated in (Fig. 3a). With those two features, our model can identify more true disease risk genes at same sample size. Further comparison showed good correlation between the prior value $\bar{\pi}_g$ informed by VBASS and real $\pi_g$ we used in simulation (Fig. 3c), indicating that VBASS could reconstruct the prior of being risk through single cell expression data. Moreover, we assessed the association between expression profile $x$ and $\pi$ via spearman correlation, the result of VBASS is close to real values (Fig. 3d). Overall, those results showed that our model can not

only reach higher statistical power on simulation data set than extTADA but also uncover the association between cell type expression profiles and disease risk.

**VBASS identified novel CHD candidate risk genes on published DNV data.** We applied VBASS to a CHD data set with DNVs from 2645 trios[13]. We used the mouse embryonic E14.5 heart bulk RNA-seq data to set gene expression rank percentile[6,7]. The estimated distribution of expression rank under null and alternative hypothesis showed most of the risk genes are enriched in rank percentile ≥75% (genes with rank percentile ≥0.75 are roughly 3 times more likely to be risk than other genes) (Fig. 4a; Table 1), consistent with previous burden analysis of de novo variants[7]. With FDR ≤0.1, we identified 49

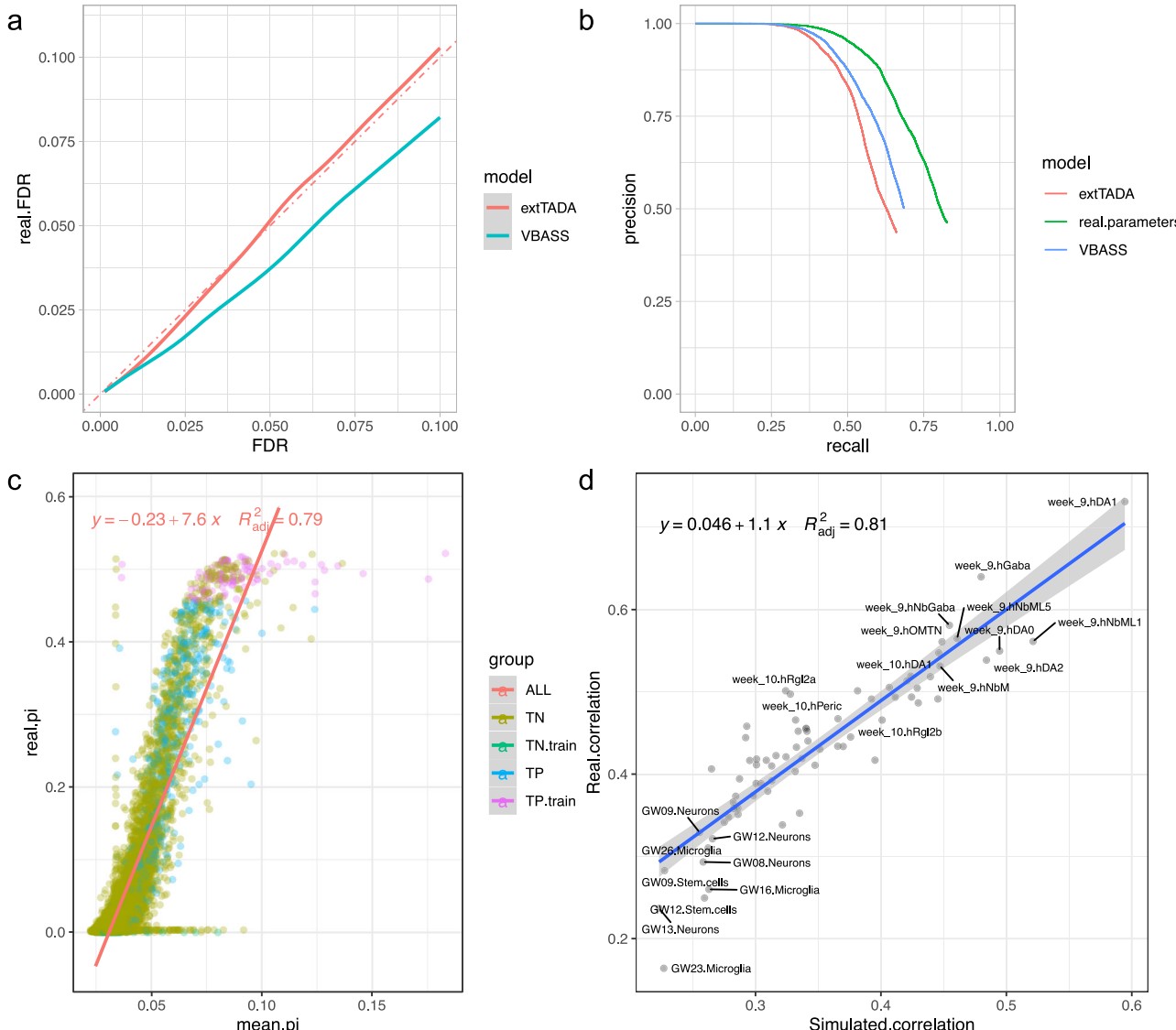

**Fig. 3 Performance of single cell version VBASS on simulation data. a** Plot of true false discovery rate (real.FDR, y axis) at different FDR cutoff (x axis) estimated by extTADA and VBASS. Genes in the training data were removed. **b** Comparison of precision recall for extTADA and VBASS, only shown for the part with FDR ≤0.5. Genes in the training data were removed. **c** Scatter plot of disease risk prior ($\pi$) that we assigned in simulation (y-axis) and informed by VBASS (x-axis). Genes were colored by labels and whether used in semi-supervised training, where TN and TP correspond to true negative and true positive, respectively. **d** Comparison of correlation between real disease risk prior and cell type expression (y-axis) versus correlation between VBASS informed prior and cell type expression (x-axis). Each dot represents a cell type. Gray shading showed confidence interval of linear regression estimated by stat_smooth function in R ggplot2 package.

candidate risk genes. In contrast, using the extTADA method, we were able to identify only 40 candidate genes (Fig. 4b, c, Table 2, Supplementary Data 4). Among the gene that only detected by VBASS, *FLT4* was reported to be a risk gene via combined analysis of de novo and inherited variants in the original paper, while *TSC1* and *FBN1* were in their curated CHD gene dataset from literature search[13,32]. *CHD4* was reported to be significantly associated with CHD in a UK CHD cohort of 1891 probands[8], while 3 (*FRYL*, *SETD5*, *KMT2C*) have both LoF and missense variants carriers, 2 (*GANAB*, *KDM5A*) have only missense variants carriers in that cohort. Furthermore, 4 (*CHD4*, *SETD5*, *KMT2C*, *FBN1*) are significantly associated with neurodevelopmental disorders[33], while 11 (*CHD4*, *FRYL*, *GANAB*, *SETD5*, *MINK1*, *ANK3*, *KMT2C*, *IQGAP1*, *TSC1*, *KDM5A*, *FBN1*) have both LoF and missense variants carriers, and 2 (*CAD*, *SLIT3*) have only

missense variants carriers in that cohort. Overall, these genes have additional genetic evidence in other cohorts and are plausible candidates. We noticed that DECO identified several other candidate risk genes when compared to VBASS (Supplementary Fig. 3a), but they showed few additional genetic evidence in the UK cohort (Supplementary Data 5). These results indicate that the assumption of VBASS is biologically sound and suggests its higher statistical power even in lower cohort size.

**VBASS identified novel ASD candidate risk genes on published DNV data.** Previous studies have shown that gene expression in multiple cell types in the brain is associated with ASD risk[22,23,34]. This is in part what motivated the design of VBASS. We obtained ASD DNV data from a recent preprint[15] that combined exome and genome data from four studies (Methods), and single cell

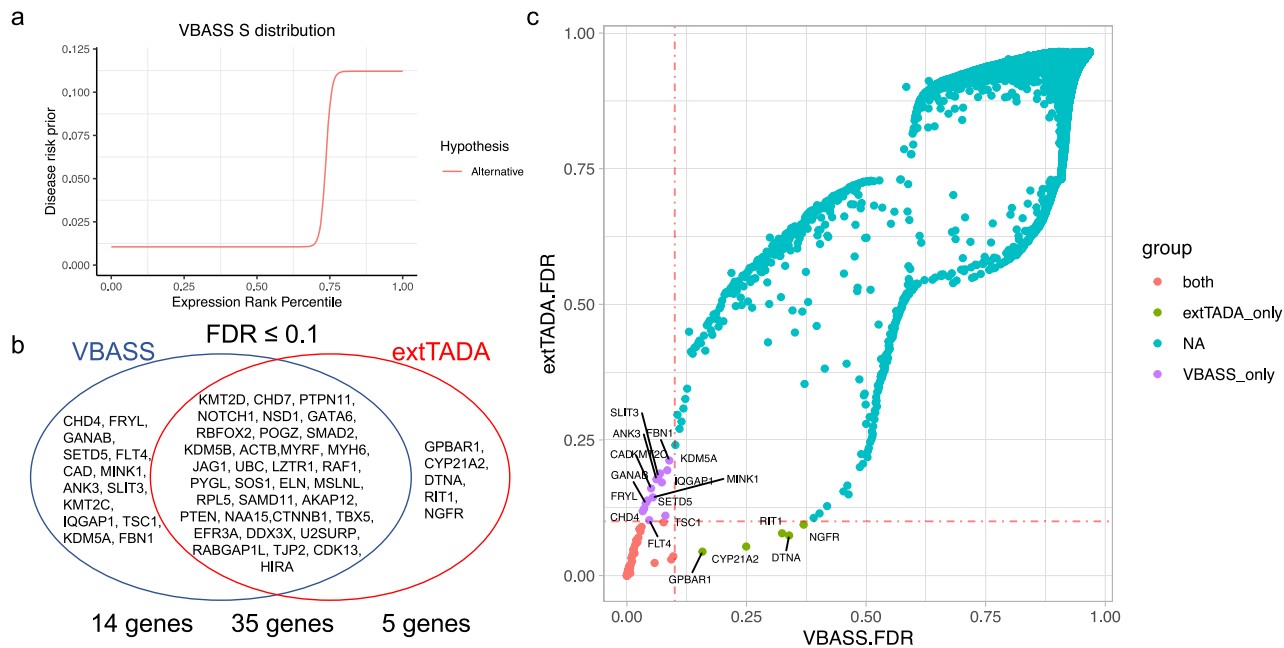

**Fig. 4 Performance comparison of VBASS and extTADA on CHD data. a** Function of disease risk prior on expression rank percentile estimated by VBASS. **b** Genes identified by VBASS and extTADA at significance level 0.1. **c** FDR of genes in extTADA (y-axis) and VBASS (x-axis), genes were colored by significance in both models (red), only in VBASS (purple) or only in extTADA (green) at significance level 0.1 (FDR ≤0.1).

**Table 1 Estimated VBASS parameters in CHD data. Mean, posterior mean; sd, standard error; 2.5% and 97.5%, confidence interval; n_eff, effective sample number in MCMC; Rhat, convergence diagnostic in MCMC.**

|  | Mean | sd | 2.50% | 97.50% | n_eff | Rhat |
|---|---|---|---|---|---|---|
| $\pi_0$ | 0.04 | 0.01 | 0.03 | 0.05 | 1845.42 | 1.00 |
| $A$ | 104.15 | 87.67 | 20.01 | 351.78 | 2263.73 | 1.00 |
| $B$ | 0.74 | 0.02 | 0.71 | 0.78 | 2136.99 | 1.00 |
| $C$ | 0.28 | 0.10 | 0.09 | 0.47 | 2093.73 | 1.00 |
| $\bar{\gamma}_{LGD}$ | 19.95 | 5.43 | 10.32 | 31.87 | 2745.30 | 1.00 |
| $\bar{\gamma}_{Dmis}$ | 11.79 | 3.60 | 5.81 | 19.36 | 3013.96 | 1.00 |
| $\bar{\beta}_{LGD}$ | 0.84 | 0.02 | 0.82 | 0.89 | 2193.48 | 1.00 |
| $\bar{\beta}_{Dmis}$ | 0.90 | 0.07 | 0.83 | 1.07 | 2144.47 | 1.00 |

**Table 2 Genes identified by VBASS but not extTADA. dn_LGD, de novo LGD variants; dn_Dmis, de novo Dmis variants.**

| Gene symbol | dn_LGD | dn_Dmis | VBASS FDR | Expression rank | extTADA FDR |
|---|---|---|---|---|---|
| *CHD4* | 0 | 3 | 0.033 | 0.990 | 0.119 |
| *FRYL* | 2 | 0 | 0.036 | 0.837 | 0.123 |
| *GANAB* | 1 | 1 | 0.039 | 0.934 | 0.133 |
| *SETD5* | 1 | 1 | 0.043 | 0.949 | 0.138 |
| *FLT4* | 2 | 0 | 0.047 | 0.734 | 0.102 |
| *CAD* | 0 | 3 | 0.051 | 0.853 | 0.161 |
| *MINK1* | 0 | 2 | 0.054 | 0.875 | 0.144 |
| *ANK3* | 2 | 0 | 0.062 | 0.948 | 0.177 |
| *SLIT3* | 1 | 1 | 0.066 | 0.860 | 0.183 |
| *KMT2C* | 1 | 2 | 0.069 | 0.792 | 0.188 |
| *IQGAP1* | 0 | 2 | 0.073 | 0.856 | 0.172 |
| *TSC1* | 1 | 1 | 0.080 | 0.728 | 0.110 |
| *KDM5A* | 0 | 2 | 0.084 | 0.859 | 0.194 |
| *FBN1* | 0 | 3 | 0.089 | 0.928 | 0.212 |

RNA-seq data of human fetal midbrain and prefrontal cortex from two publications ("Methods")[26,27]. We applied VBASS and extTADA to the full ASD data set with 16,616 trios. VBASS identified 122 genes with PPA above 0.8 (Supplementary Data 6). To compare the performance in identification of novel candidate risk genes, we removed the known risk genes used in training and calculate Bayesian FDR of all other genes with VBASS and extTADA (Methods). Then we compared the candidate genes identified by VBASS and extTADA at significance level 0.05 and 0.1 (FDR ≤0.05 and FDR ≤0.1 respectively). At significance level 0.05, VBASS identified 51 genes (Supplementary Data 7), among which 5 were not identified as candidates by extTADA (Fig. 5a, Supplementary Data 7). Among the 5 genes, 2 (*DLG4*, *PAX5*) were reported to be risk genes in SFARI[35] data base (release 2021 Q4) with score of 1 while not in our training gene list. *METTL23* is a transcriptional partner of GABPA and essential for human recognition[36], and disruption of *METTL23* was reported to cause mild autosomal recessive intellectual disability[37]. *ATF4* was reported to have significant altered expression in the middle frontal gyrus of ASD subjects[38]. At significance level 0.1, VBASS identified 75 genes (Supplementary Data 7), where 6 were not identified by extTADA (Fig. 5a, Supplementary Data 7). Among the 6 genes, 2 (*ZMYND8 CASZ1*) were scored 1 in SFARI data base and *CMPK2* was scored 3. *LMTK3* was reported to cause behavioral abnormalities such as locomotor hyperactivity and reduced anxiety in mice knock-out models[39,40]. Furthermore, 7 out of the 11 genes identified only by VBASS (*DLG4, METTL23, SPRY2, LMTK3, PFN2, CASZ1, ZMYND8*) have additional genetic evidence in related cohorts[33]. There were six genes (*CCDC40, FUBP3, PRKAR1B, SIN3A, ITGB5, PMM2*) identified only by extTADA but not by VBASS, likely because of their low detection rates or co-expression strength with other candidates in the single cell datasets. Compared with VBASS, DECO identified a similar number of additional candidate genes. Most of these genes (25 out of 35) are from a single gene set (gene ontology term "Positive regulation of gene expression"). (Supplementary Fig. 3b, Supplementary Data 8).

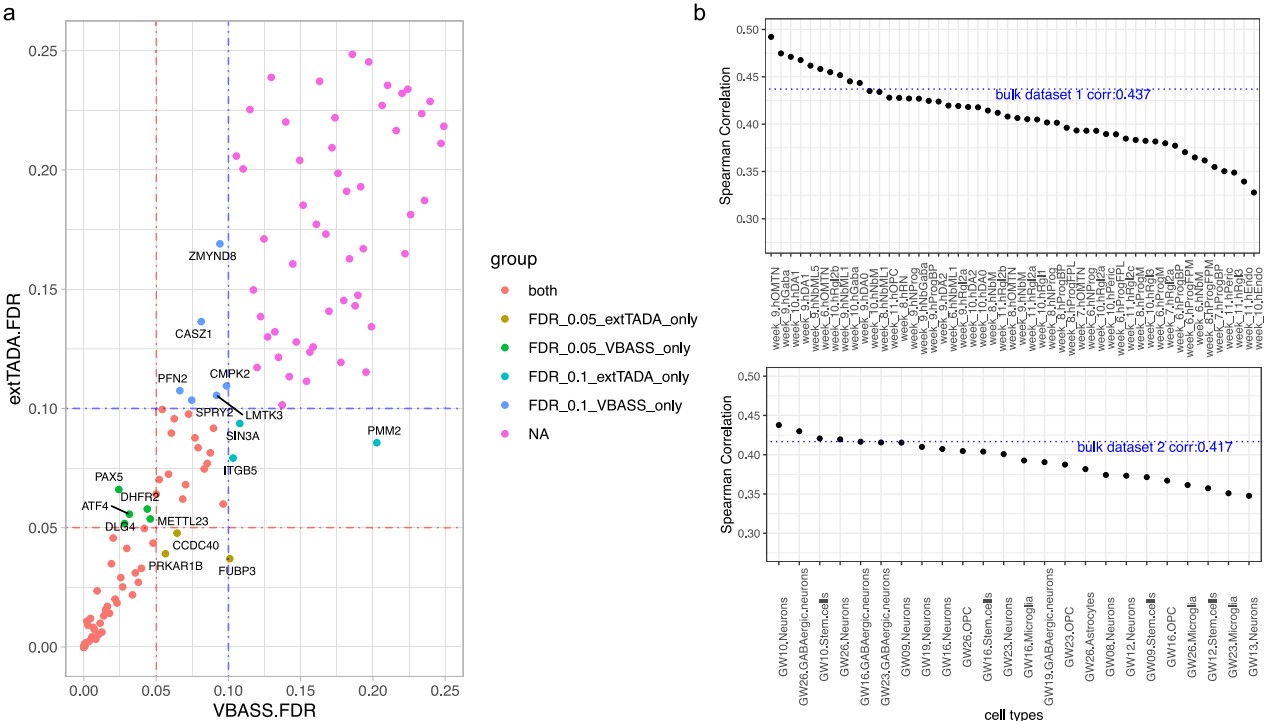

**Fig. 5 Performance comparison of VBASS and extTADA on ASD data. a** FDR of genes in extTADA (y-axis) or VBASS (x-axis). **b** Spearman correlation between cell type expression and disease risk prior ($\pi$). The cell types from two single cell data sets were separated and ordered by correlation with $\pi$, respectively.

To further investigate whether single cell data is more informative than bulk data in VBASS, we compared the results with bulk version VBASS predictions using pseudo bulk expression data that curated from the single cell data ("Methods"). As expected, in complex conditions like autism, bulk version VBASS failed to detect some well-known risk genes like *TSC2, CSNK2B, SPRY2*, and *ZMYND8* (Supplementary Data 7), because it does not have the information to model the correlation between cell type heterogeneity and disease risk. Finally, we studied what are the cell types that associated the most with disease risk. VBASS is designed to assign high risk prior to genes that highly expression in certain disease related cell types while low in other cell types, as shown in (Supplementary Fig. 4a, Supplementary Fig. 4b), risk genes tend to express highly in both progenitor and mature neuronal cell types (genes on the top, Supplementary Fig. 4b) or only a few specific neurons (genes in the middle, Supplementary Fig. 4b). VBASS quantified this co-expression pattern by non-linear functions (Supplementary Fig. 5a), and we can show the contribution of each cell type in the non-linear function qualitatively by PCA analysis (Supplementary Fig. 5b) or quantitively by Spearman correlation (Fig. 5b). We notice that oculomotor / trochlear nucleus (hOMTN), GABAergic neurons (hGaba) and dopaminergic neurons (hDA1) in gestation week 9-10 are more associated with autism risk, while microglia cells and endothelial cells (hEndo) are less associated with autism risk (Fig. 5b). This observation is consistent with previous evidence of abnormalities in GABAergic neurons and synapses in neurodevelopmental disorders characterized by a shared symptomatology of ASD symptoms[41], while reductions in GABA have been reported in several brain regions in children with ASD[42,43]. There were also evidences that dopaminergic dysfunctions were associate with autistic-like behavior[44,45].

## Discussion

In this study, we described VBASS for identification of candidate risk genes by joint analysis of de novo variants of cases and gene expression profile of normal samples. The core idea of the method is that prior probability of a gene increase disease risk is a function of expression profile in relevant cell types, and that we can estimate the parameters of the function from the data in an empirical Bayesian framework. For bulk RNA-seq data, we set the function to be a sigmoid function with three parameters. For scRNA-seq data, we use deep neural networks to approximate the function and learn the contribution of cell types jointly with genetic data. Using simulation, we showed that VBASS have accurate error rate control and better statistical power than existing method extTADA under both scenarios.

We applied VBASS to a published CHD DNV data set and estimated that high-expression genes are approximately 3 times more likely to be risk genes than low-expression genes in developing heart. We identified 14 more candidate risk genes, 6 of which have additional support in independent cohorts. We applied VBASS to a published ASD DNV data set and identified 5 and 6 more candidate genes at significance level 0.05 and 0.1 respectively, 8 of them have literature support or additional genetic evidence in neurodevelopmental disorders. Moreover, we showed that gene expression profiles of GABAergic neurons and dopaminergic neurons during gestation week 9-10 are strongly associated with autism risk, indicating their potential roles in neural circuits formation.

VBASS is based on the biological hypothesis that gene expression level in relevant cell or tissue types informs the plausibility of being a disease risk gene. The bulk-RNA seq version is optimized for a single expression profile that is informative of disease risk, such as bulk RNA sequencing data for congenital heart disease. The single cell RNA-seq version is optimized for the

conditions in which multiple cell types and time points are associated with disease risk. Furthermore, the main assumption of VBASS is that the disease risk prior is a function of gene expression levels in cell types or tissues. The validity of this assumption can be examined qualitatively using PCA of the gene expression matrix. If the assumption holds, we should observe a partial separation of known risk genes or potential risk genes (such as de novo LGD variants in cases with developmental disorders) from other genes by one of first few principal components.

Generally, single cell version is more informative than bulk version, especially for heterogeneous conditions like autism in which different risk genes may have critical functions in different cell types. One alternative approach to improve power based on informative non-genetic data is to calculate p-values for each gene using genetic data and then optimize FDR estimation using non-genetic data as covariates[46–48], While it is a generalizable approach, these methods require $p$ values to have proper distributions (uniform) under the null. In the analysis of de novo or ultra-rare variants, the data is usually too sparse to support a proper distribution of p-values under the null. VBASS does not have this limitation. Other methods that rely on information like curated gene sets or pathways can increase power for genes included in the pre-defined sets.

A limitation of VBASS is that it only estimates the association of cell types with disease risk. It is not designed to answer questions about whether a certain cell type confers causality in the diseases caused by risk variants. Additionally, the performance of VBASS is partially determined by how well the expression data captures true expression states of genes. In this study, we used average expression of genes in cells within a cell type inferred from single cell data. This approach has limitations in representing rare and transient cellular states. More advanced representation, like RNA velocity[49,50], together with more comprehensive measurements of cell types may improve the model.

## Methods

**The probabilistic model of VBASS.** VBASS is a Bayesian mixture model with learnable priors. VBASS assumes the number of genetic variants of interest (LGD or Dmis de novo variants) in the gene $d_{gv}$ are drawn independently through this generative process, given $M_{gv}$ being the aggregated background mutation rate for variant type $v$ in gene $g$ and $x_g$ being the cell type specific gene expression profiles in gene $g$:

$$
\begin{aligned}
\pi_g &= f_E(x_g) \\
y_g &\sim Bernoulli(\pi_g) \\
k_{gv} &= \begin{cases} \overline{k_v} & \text{if } y_g = 1 \\ 1 & \text{otherwise} \end{cases} \\
\theta_{gv} &= \begin{cases} \overline{\theta_v} & \text{if } y_g = 1 \\ 1 & \text{otherwise} \end{cases} \\
\gamma_{gv} &\sim \begin{cases} Gamma\left(k_{gv}, \theta_{gv}\right) & \text{if } y_g = 1 \\ 1 & \text{otherwise} \end{cases} \\
d_{gv} &\sim \begin{cases} Poisson\left(\gamma_{gv} * M_{gv}\right) & \text{if } y_g = 1 \\ Poisson\left(M_{gv}\right) & \text{otherwise} \end{cases}
\end{aligned} \tag{1}
$$

$\pi_g$ is a gene specific prior probability of being disease risk. $y_g$ is a binary random variable that indicates the risk status of a gene, which follows a Bernoulli distribution of $\pi_g$. We use neural network $f_E$ to infer $\pi_g$ from gene expression data $x_g$ with a penalty term of Kullback–Leibler divergence over average proportion of disease risk gene, $\bar{\pi}$. (Supplementary Fig. 1). This penalty term could be replaced by a cross-entropy loss term if the label of gene is known, making it possible for semi-supervised training[51].

By default, we used a 32-dim encoding module, followed by a 2-dim sampler module for $\pi_g$. Each module consists of a linear layer followed by ELU activation

and layer normalization layers. We apply the same reparameterization trick as conventional variational autoencoders in $f_E$ with Bernoulli sampler[52].

$\gamma_{gv}$ is a random variable that denotes the enrichment rate of damage variant $v$ in the patient cohort, which is also known as the relative risk of this gene. $\gamma_{gv}$ is drawn independently through Gamma distribution $p(\gamma_{gv}|k_{gv}, \theta_{gv})$. $k_{gv}, \theta_{gv}$ are conditioned on $y_g$, under null they are equal to 1 while under alternative, they are equal to $\overline{k_v}$ and $\overline{\theta_v}$, respectively. We assume $\overline{k_v}$ and $\overline{\theta_v}$ are shared across all disease risk genes to reduce the number of parameters, similar to the assumption in TADA and extTADA[12,30]..

The loss function is given by the evidence lower bound (ELBO),

$$ ELBO = -KL[q(y|x)||p(y)] - \mathbb{E}_{q(y|x)}\log(p(d|y)) \tag{2} $$

The KL penalty term regularized the gene-specific prior $\pi$ by the hyperparameter $\bar{\pi}$, which reflects the average proportion of risk genes:

$$ KL[q(y|x)||p(y)] = KL[Bernoulli(y|\pi;f_E(x))||Bernoulli(y|\bar{\pi})] \tag{3} $$

The expectation term quantified the log likelihood of $d$ conditioned on $y$ integrated on the distributions parameterized by $\pi$:

$$ \mathbb{E}_{q(y|x)}\log(p(d|y)) = \int Poisson(d|\gamma * M; \overline{k_v}, \overline{\theta_v}) * Bernoulli(y|\pi; f_E(x))dy \tag{4} $$

$f_E, \overline{k_v}, \overline{\theta_v}$ are the parameters to learn, we use stochastic gradient decent to estimate them. The estimated parameters were used to calculate the posterior probability of association (PPA) for each gene being risk or not:

$$ PPA = \frac{\pi_g * GammaPoisson\left(d_{gv}|\overline{k_v}, \overline{\theta_v}, M_{gv}\right)}{\pi_g * GammaPoisson\left(d_{gv}|\overline{k_v}, \overline{\theta_v}, M_{gv}\right) + \left(1 - \pi_g\right) * Poisson\left(d_{gv}|M_{gv}\right)} \tag{5} $$

For conditions where gene expression data $x_g$ is a scalar, i.e., bulk RNA-seq data or average expression data of a certain cell type in scRNA-seq data, we could rewrite $f_E$ as a function with sigmoid shape, corresponding to a linear transformation with sigmoid activation:

$$
\begin{aligned}
f_E(x_g) &= \bar{\pi} * Sigmoid(x_g|A, B, C) \\
Sigmoid(x_g|A, B, C) &= C + \frac{L}{1 + A * \exp(-x_g + B)} \\
L &= (1 - C) * \frac{A}{\log(\exp(A) + \exp(A * B)) - \log(\exp(A * B) + 1)}
\end{aligned} \tag{6}
$$

while the other parts of the model remain the same.

Given PPA of all genes, we calculate Bayesian false discovery rate (FDR) by estimated false discovery proportion following the method described in He et al., 2013[12]:

$$ FDR_k = \frac{\sum_{i=1}^k \left(1 - PPA_i\right)}{k} \tag{7} $$

where $i$ is the rank index of genes (start with highest PPA), and $FDR_k$ is the estimated FDR of the gene ranked at $k$.

**Parameter inference for VBASS.** The parameters of VBASS could be inferenced with either unsupervised or semi-supervised training. For the bulk version, there are only six parameters to be estimated, $\bar{\pi}, \overline{\theta_v}, \overline{k_v}, A, B, C$, which is possible for complete unsupervised training via MCMC. In practice, we used Rstan package with 4 chains and 2000 iterations. For the neural network version, we recommend two settings to avoid converging issues and keep good control of false discovery rate. First is setting the hyperparameter of KL penalty to the observed average proportion of risk genes, in practice we recommend estimating that using extTADA or running VBASS without gene expression data. We showed that either higher or lower KL penalty will result in improper false discovery control (Supplementary Fig. 6). Second is to train in a semi-supervised manner. In practice, we trained the model with two training steps. First, we pre-trained our model using known risk genes labeled as positives and genes that harbor LGD variants in control cohort as negatives, replacing the Bernoulli KL penalty with cross-entropy loss[51]. The known risk genes (59 in total) were randomly picked from SFARI[35] (release 2021 Q4) scored 1 genes, while negative controls (86 in total) were picked from genes with LGD variants in a control cohort[14] (Supplementary Data 1). During pre-training we set large learning rate to make the model converge faster. The parameters estimated from pre-training were then used as initial values in the second step, unsupervised training, which uses all genes without labels with reduced learning rate after each epoch. In practice, we used 50 epochs of semi-supervised training and 60 epochs of unsupervised training. After training, we calculated PPA for all genes using the estimated parameters. For the simulation dataset, we estimated FDR on all genes to measure the statistical power. For the real dataset, we removed the known risk genes selected as positives in training, which should have very large PPA due to their large effect sizes (Supplementary Fig. 7), when we estimate FDR to identify candidate novel risk genes.

**De novo variants (DNV) and gene expression data.** We obtained DNV data sets from a publication on congenital heart disease (CHD)[13] of 2645 parent-offspring

trios (Supplementary Data 2) and a preprint on autism spectrum disorder (ASD)[15] of 16,616 trios (Supplementary Data 3). The subjects in first data set (CHD) were recruited to the Congenital Heart Disease Network Study of the Pediatric Cardiac Genomics Consortium[53] (CHD GENES: ClinicalTrials.gov identifier NCT01196182). The institutional review boards of Boston's Children's Hospital, Brigham and Women's Hospital, Great Ormond Street Hospital, Children's Hospital of Los Angeles, Children's Hospital of Philadelphia, Columbia University Medical Center, Icahn School of Medicine at Mount Sinai, Rochester School of Medicine and Dentistry, Steven and Alexandra Cohen Children's Medical Center of New York, and Yale School of Medicine approved the protocols. All subjects or their parents provided informed consent[13]. The second data set (ASD) is a combined data set from exome or whole genome sequencing data of the SPARK consortium[54], Simons Simplex Collection[55], Autism Sequencing Consortium[56], and MSSNG[57]. All participants were recruited to SPARK under a centralized institutional review board (IRB) protocol (Western IRB Protocol no. 20151664). All participants provided written informed consent to take part in the studies. Written informed consent was obtained from all legal guardians or parents for all participants aged 18 and younger and for all participants aged 18 and older who have a legal guardian[15]. For CHD analysis using bulk version of VBASS, we input the gene expression rank value, ranged from 0-1, for each gene. The gene expression rank was based on bulk RNA-seq data of mouse developing heart at E14.5, inspired from previous publications[6,7]. For ASD analysis using single cell version of VBASS, w~~We~~ obtained single cell RNA-seq data of human fetal midbrain and prefrontal cortex from two publications[26,27]. We used the combination of developmental time and cell ontology annotations as described in the two publications to define cell types. Small clusters of cell types with less than 10 single cells were removed. For each gene ~~and cell type~~, we calculate the proportion of cells that express the gene in each timepoint specific cell types, ranged from 0-1, as the input vector to VBASS. To run the bulk version of VBASS on ASD, we generated pseudo bulk gene expression value, ranged from 0-1, as the proportion of cells that express the gene in all cell types and input to bulk version of VBASS, while other settings remain as default. As we are using single cell data from two publications, we generated the pseudo bulk data and run bulk version of VBASS separately to avoid the impact of batch effect.

**Annotation of de novo variants and background mutation rate calculation**. We used ANNOVAR[58] and VEP[59] to annotate variants, protein-coding consequences, and predicted damaging scores for missense variants. We classified variants as LGD (likely gene disrupting, including frameshift, stop gained/lost, start lost, splice acceptor/donor), Dmis (Damage missense variants, defined by REVEL[60] score ≥0.5), missense, or synonymous. For each variant type, we calculated the expected number of variants based on a background mutation rate model[7,61] given the sample size. In-frame deletions/insertions (multiple of 3 nucleotides) and other splice region variants were excluded in the following analysis. Variants in olfactory receptor genes, HLA genes or MUC gene family were excluded in further analysis.

**Generation of simulation datasets**. We simulated two datasets to test VBASS's performance with bulk and scRNA-seq datasets, respectively. For the first scenario, we first estimate the parameters based on real dataset and then used the estimated hyperparameters to generate the simulated dataset based on the Bayesian mixture model. Specifically, we randomly assigned 3.7% of genes as risk gene, then we drew the covariates (gene expression rank) of risk genes from the sigmoid distribution function. The de novo damage variants were drawn from Gamma-Poisson distribution with relative risk of 20 and 12 for LoF and Dmis, respectively. For non-risk genes, we drew covariates from a uniform distribution and de novo variants from Poisson distribution. We did the simulation under different sample sizes ranging from 2645 to 20,000. For each sample size setting, we simulated 100 datasets and fit both models on each simulated dataset independently to estimate the hyperparameters, which were used to calculate the posterior probability of association (PPA) and then a Bayesian false discovery rate (FDR) by false discovery proportion implied by it. We performed single-tail Poisson tests independently on each simulated dataset to show the baseline statistical power, where the FDR were calculated by the Benjamini-Hochberg (BH) method.

For the second scenario, the simulation is based on real single cell dataset, where we created a non-linear function that maps cell-type specific expression to prior of being risk with following steps. First, we did a singular value decomposition (SVD) on the expression data of 59 known ASD risk genes (picked randomly from SFARI[35] scored 1 genes) and 86 negative control genes (picked randomly from genes with LGD variants in control cohort[14]) (Supplementary Data 3). Next, we fit a logistic regression model with elastic net penalty on the eigen vectors that explain 95% of the variance. The regression model was applied to all other genes and the output probabilities were squared and scaled to have an average of 3.2%, which matches the average proportion of risk genes estimated from extTADA model. This value served as a simulated prior of being risk, from which disease risk genes were randomly sampled (~600 risk genes). The de novo damage variants were drawn from Gamma-Poisson distribution for disease risk genes while Poisson distribution for non-risk genes with same sample size and relative risk as in real ASD dataset. We performed the simulation 50 times with same simulated prior and disease risk genes, then estimated the hyperparameters and calculate the PPA and Bayesian false discovery rate (FDR) independently on

each simulated dataset for both models. For VBASS, we randomly selected ~100 simulated risk genes and ~300 non-risk genes as label for the semi-supervised training. Those genes were excluded for comparison.

**Comparison with DECO on CHD and ASD**. We performed DECO[21] on real CHD and ASD de novo variants dataset following the recommended protocols. We downloaded 21604 gene sets (≥5 genes) from MSigDB[62] dataset (Version March 2020). We ran DECO for each gene set and selected the gene set with lowest p-value for comparison with VBASS. The gene sets were GO_HEART_DEVE-LOPMENT and GO_POSITIVE_REGULATION_OF_GENE_EXPRESSION for CHD and ASD, respectively. We calculated the Bayesian FDR using PPA for DECO in the same way as VBASS.

**Reporting summary**. Further information on research design is available in the Nature Portfolio Reporting Summary linked to this article.

## Data availability
All data sets (de novo variants and gene expression data) were obtained from publications and available from the corresponding publications. The accession number of the single cell expression data used in this paper are GSE76381 and GSE104276. We also included the annotated de novo variants in the supplementary data 2 (CHD) and supplementary data 3 (ASD). All source data underlying the figures were provided in Supplementary Data 9.

## Code availability
VBASS is available on GitHub: https://github.com/ShenLab/VBASS and Zenodo: https://doi.org/10.5281/zenodo.8018227[63].

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

## Acknowledgements

This work was supported by NIH grants R01GM120609, R03HL147197, R03HL161595, P01HD093363, R35GM149527, and U01HL153009, and Simons Foundation Autism Research Initiative (SIMONS606450). We would like to thank Dr. Wendy Chung, Dr. David Knowles, Dr. Nicholas Tatonetti, Dr. Haicang Zhang, Dr. Xueya Zhou, Dr. Xiao Fan, Yige Zhao, Joseph Obiajulu, Xi Fu, and members of Shen lab for helpful discussions.

## Author contributions

Conceptualization, Y.S.; Methodology, G.Z. and Y.S.; Software, G.Z.; Investigation, G.Z., Y.A.C. and Y.S.; Writing – Original Draft, G.Z. and Y.S.; Writing – Review & Editing, G.Z., Y.A.C. and Y.S.; Supervision, Y.S.; Funding Acquisition, Y.S.

## Competing interests

The authors declare no competing interests.
