## [Peer Review File · Communications Biology]

Reviewers' comments:

Reviewer #1 (Remarks to the Author):

In this manuscript Zhong et al. present a novel computational method using a Bayesian framework to leverage gene expression data for the prioritization of underpowered rare variant genetic association signals. These type of methodologies are vital as the sample sizes required to identify the genetic causes of highly pleiotropic and/or genetically heterogeneous diseases (such as CHD, ASD, SCZ, ...) are simply prohibitive from a sample recruitment as well as an experimental perspective. The combination of publicly available large-scale, tissue-specific expression atlases make the proposed method highly relevant and of interest to many ongoing large-scale genetic association studies. The adoption of the tool by the community, and thus its impact, will depend not only on its ability to successfully prioritise candidate genes with weak association signals, but also on its useability. Although I think the authors make a compelling argument for the method, I do have some questions/remarks:

Major:

- 1) The description of the model and its different components can be quite abstract and challenging for the non-technical reader, which in the end will be its end-user. Figure 1 doesn't really help much in understanding how the model works. Adding a simplified description of the model with easily understandable descriptions of what each of the inputs and the learned parameters represent (maybe with numerical examples of what a pathogenic gene vs a non-pathogenic gene would look like) will likely help adoption of the tool and improve the manuscript.
- 2) In Figure 3C, the estimated π 's for genes which were included in training seem generally higher than those in testing. Could this be due to overfitting? Can the authors comment about the generalization of the method with respect of the input gene expression dataset and the training set?
- 3) How would you know that the correlation between disease risk and cell type is significant (e.g. when the authors state in the discussion that there is "strong association" between certain neuron subtypes and gene-wise disease risk)? Wouldn't you expect particular cell types to be clearly relevant and thus be well beyond the "baseline" correlation of unrelated cell types? It's also unclear if the model assumes that all disease risk genes should be expressed in the same cell type, which is unlikely to be the case. How would cell type heterogeneity affect the presented results?
- 4) The presented improvements in error rate control/statistical power, how do they translate to real life scenarios? I think it would help the reader to conceptualise what for example the improvement in recall in VBASS in Figure 3B really means.
- 5) The authors assume that all disease genes belong to the same effect size distribution parametrized by 2 shared hyperparameters. Is this assumption valid based on known disease genes?
- 6) In the KL penalty term how is the π -hyperparameter determined for the average proportion of disease risk genes? Does the choice of this parameter influence the results significantly?

Minor:

- 1) Given that the authors compare their tool to exTADA, it would be useful to include a section of the introduction which describes the differences/similarities between the two methods
- 2) Currently the authors collapse the single-cell data per cell type into a vector which is then used as input into the NN to predict π , instead of using the complete cell-wise count matrix which might allow the model to take into account within-cluster transcriptional heterogeneity. Wouldn't this additional information be useful for the prediction?
- 3) It's not clear to me what the authors mean by the last few sentences of page 6 when they discuss this p-value based approach.

Reviewer #2 (Remarks to the Author):

Summary:

This paper presented a Bayesian mixture model called VBASS with priors trained from single cell

gene expression by the neural network to model the number of genetic variants. The author indicated that their method can be applied to both single cell data and bulk level data by replacing the neural network with a sigmoid function. The paper compared VBASS with a published state-of-art computational method extTADA on two types of datasets, which are simulated datasets and published real datasets. The author showed the model's statistical power by increasing sample size/gene size and showed its ability to identify novel risk genes by comparing candidate risk genes found by VBASS and exTADA and validating based on literature search.

Major comments:

1. Something I found unclear in the paper is how exactly single-cell data as opposed to bulk is helping the model. It is clear that the expression profile of a given gene across the cell types is used to determine the prior probability of being a risk gene, but the authors don't elaborate on how this benefits the model. For example, would a gene with a high level of expression across all cell types be considered more important than one that is highly variable or lowly expressed? I think clarifying what kind of profiles are expected to give a higher prior (or at least what profiles they found gave a higher prior) would help here.
2. The authors mention that the method only finds the correlation between cell types and risk genes and does not infer causality. It would be helpful if the authors discussed what changes they would have to make to enable to model to infer that. This is particularly important as being able to identify the relevant cell types would help justify the use of single cell data. For example, could they use a cell type-specific prior?
3. It would have been interesting if the authors had compared the bulk VBASS method to the single-cell approach. This could have helped show if /how the use of single-cell data allows for the identification of more risk genes and/or narrows down what cell those genes are relevant in (i.e run analysis with pancreas bulk vs. single-cell pancreas).
4. The authors compared their method to extTADA which seems to be the gold standard for integrating scRNA-seq with DNV data for risk gene discovery. Only having one method used for comparison limits our ability to grasp how well the method performs as there are few reference points. Assuming that this was due to the lack of single cell approaches, the authors could see if there are bulk RNA methods, they could compare to bulk VBASS to have more benchmarking comparisons. Given the similarity between the bulk and single-cell VBASS, this could help prove the reliability of their model.
5. In the section "...identified novel CHD candidate risk genes on published DNV data", the author compared VBASS with the original TADA method. What is the intuition here to compare with the original TADA instead of extTADA?
6. They have compared the results of their method with only one another approach. It would be good to compare with a few more approaches, if possible, such as mTADA, EncoreDNM.
7. The author showed that they can identify novel ASD risk genes on published data by removing known risk genes before training the model. How is VBASS's performance in identifying known risk genes? It would be more comprehensive if the author could add a section stating the model's ability to identify known risk genes on the published dataset.
8. Since the method can be applied on both single cell RNAseq and bulk RNAseq datasets, it would be nice to compare the results from a case where both scRNAseq and bulk RNAseq data were drawn from the same sample. This would provide an idea of how important it is / or not to have access to the high-resolution, high-cost scRNAseq datasets for such analyses.

Minor comments:

1. Grammar and Spelling: There were a few errors across the manuscript. I did not cite them here as I am not sure it is necessary for this stage of the review but can send them if required.
2. The introduction should discuss the different types of gene variants and their impact, to provide a better context of the biological research area
3. The introduction should outline and discuss the other methods developed in this area, to provide a better context of the existing methods.

Reviewer #3 (Remarks to the Author):

Zhong and co-author introduce VBASS, a Gamma-Poisson model to predict gene disease association from GWAS data. A novel aspect of the work is the use of gene expression data (from standard or single cell sequencing) as a prior for the Gamma-Poisson model. This is overall a sound and relevant method. They use simulated data to evaluate the statistical power of the approach and to compare it with a single competing method (extTADA). They also present the application of the method in two distinct real data sets (on coronary heart disease and Autism spectrum disorder), where they show that genes only detected by their method are of potential functional relevance. This is an overall interesting study, it fails short in a few important details.

1. The proposed model is quite intricate and explained with typical graph model representation in figure 1. Given that the *Comms. Biology* has a more general readership, authors should make an effort to make figure 1 more conceptual, i.e. explanation of the parameters / input data / random variables.

2. There are not much details on how single cell gene expression is used. For the DNV study, I could only find a simple passage "single cell RNA-seq data of human fetal midbrain and prefrontal cortex from two publications.". How are the expression vectors x_g parametrized? All individual cells independently of the cell type? How biased would the model be towards more abundant cell types? How is the data processed? Authors should expand this explanation and have a materials section dedicated to these. The same applies for the CHD analysis.

3. The manuscript and benchmarking only mentioned one alternative competing method. There are several other similar methods in the literature (see below for a few examples), which are not cited. Authors need to put all competing methods in perspective and evaluate relevant methods in the benchmarking analysis.

<https://www.ncbi.nlm.nih.gov/pmc/articles/PMC5738153/>

<https://journals.plos.org/plosgenetics/article?id=10.1371/journal.pgen.1009849>

<https://academic.oup.com/bib/article/22/5/bbab067/6206366>

4. In the same line, authors should evaluate a variant of the VBASS, where no gene prior information is provided.

Response to reviewers' comments:

Reviewer #1:

In this manuscript Zhong et al. present a novel computational method using a Bayesian framework to leverage gene expression data for the prioritization of underpowered rare variant genetic association signals. These types of methodologies are vital as the sample sizes required to identify the genetic causes of highly pleiotropic and/or genetically heterogeneous diseases (such as CHD, ASD, SCZ, ...) are simply prohibitive from a sample recruitment as well as an experimental perspective. The combination of publicly available large-scale, tissue-specific expression atlases make the proposed method highly relevant and of interest to many ongoing large-scale genetic association studies. The adoption of the tool by the community, and thus its impact, will depend not only on its ability to successfully prioritize candidate genes with weak association signals, but also on its useability. Although I think the authors make a compelling argument for the method, I do have some questions/remarks:

Major:

1) The description of the model and its different components can be quite abstract and challenging for the non-technical reader, which in the end will be its end-user. Figure 1 doesn't really help much in understanding how the model works. Adding a simplified description of the model with easily understandable descriptions of what each of the inputs and the learned parameters represent (maybe with numerical examples of what a pathogenic gene vs a non-pathogenic gene would look like) will likely help adoption of the tool and improve the manuscript.

We thank the reviewer for this suggestion. We made a new Figure 1 as suggested, showing the model input/output and key parameters. We also included numerical examples of model input and output of disease risk and non-risk genes. We converted the previous Figure 1 to Supplementary Figure 1 as it demonstrated the graphic model where nodes represent parameters/variables and arrows indicate dependency. We revised the figure description to make it easier to understand.

2) In Figure 3C, the estimated π 's for genes which were included in training seem generally higher than those in testing. Could this be due to overfitting? Can the authors comment about the generalization of the method with respect of the input gene expression dataset and the training set?

Figure 3C is based on simulations. During simulation, as described in Methods part "Generation of Simulation datasets", "we created a non-linear function that maps cell-type specific expression to prior of being risk", and "disease risk genes were randomly sampled" from this prior. We selected disease risk genes with highest risk priors as positive points in semi-supervised training during simulation, which was reflected in Figure 3C that those points have large "real. π " value. The simulation proves that, if the disease risk prior is associated with input gene expression, even in the format of a non-linear function, VBASS could learn it. We note that risk prior is a latent variable in the model, and that the accuracy of the method is ultimately evaluated by error rate control in Figure 3A.

In the revision, we added a paragraph in the Discussion section about generalization of the method in real applications:

the underlying assumption of VBASS is that the disease risk prior is correlated with the input gene expression dataset. The validity of this assumption can be checked qualitatively using PCA of the gene expression matrix. If the assumption holds, we should observe a separation of known risk genes or potential risk genes (such as high enrichment of LGD variants in cases) from other genes by one of first few principal components, then we could use those genes as training set.”

3) How would you know that the correlation between disease risk and cell type is significant (e.g. when the authors state in the discussion that there is “strong association” between certain neuron subtypes and gene-wise disease risk)? Wouldn’t you expect particular cell types to be clearly relevant and thus be well beyond the “baseline” correlation of unrelated cell types? It’s also unclear if the model assumes that all disease risk genes should be expressed in the same cell type, which is unlikely to be the case. How would cell type heterogeneity affect the presented results?

“Strong association” means the spearman correlation of those cell types to π is relatively higher than the other cell types. We revised Figure 5B with a dash line showing the spearman correlation between disease risk prior and the bulk expression profile.

We didn’t assume that all disease risk genes are expressed in the same cell type. In the revision, we added a new Supplementary Figure 3 to show the hierarchical clustering of cell types and genes used in training and significant genes in testing. We notice that overall gene expression itself is a good but weak proxy of risk genes: risk genes tend to express highly in both stem and mature neuronal cell types (genes on the top) or a few specific neurons (genes in the middle).

Supplementary Figure 3. Heatmap plot of gene expression profiles for some selected genes. We color the genes we used in semi-supervised training as red (known disease risk) and blue (known disease non-risk) on the left, correspondingly (marked as “training.label”). We color the significant (FDR ≤ 0.1) genes and non-significant genes (FDR > 0.1) as red and blue (marked as “posterior.label”). We plot the log disease risk prior inferred by VBASS on the right (marked as “log_pi”). The genes were hierarchically clustered by significance of FDR and expression values.

Many cell types have correlated expression patterns, as shown in a new figure Supplementary Figure 4 (below) based on PCA of the cell-type by gene expression profile. PC1 reflects an average expression of a gene across all cell types, PC2 reflects the heterogeneity of two single cell datasets, PC3 reflects the difference of differentiated neurons, progenitor cells, and non-neuronal cells. In VBASS, we do not make assumption about which cell type(s) are important or risk genes must be highly expressed in certain cell types. Instead, the main idea of the method is that it will learn a (non-linear) function from cell-type-specific expression to risk prior based

on observed genetic data. We added this part to Supplementary Figure 4 and Section “VBASS identified novel ASD candidate risk genes on published DNV data”:

VBASS is designed to assign high risk prior to genes that highly expression in certain disease related cell types while low in other cell types, as shown in (Supplementary Figure 3), risk genes tend to express highly in both progenitor and mature neuronal cell types (genes on the top) or only a few specific neurons (genes in the middle). VBASS quantified this co-expression pattern by non-linear functions (Supplementary Figure 4A), and we can show the contribution of each cell type in the non-linear function qualitatively by PCA analysis (Supplementary Figure 4B) or quantitatively by Spearman correlation (Figure 5B).

Supplementary Figure 4. The non-linear function between disease risk prior and cell type specific expressions. A) Visualization of the non-linear function of disease risk prior ($\log(\pi)$) on principle components (PCs) of gene expression. Each dot is a gene, x-axis showed its value of each PC, which is a linear combination of the co-expressed cell types, y-axis showed its log disease risk prior. The blue line showed a local regression line that approximately visualize the non-linear function. B) PCA plot of gene expression. Each dot is a gene, each arrow showed the importance of each cell type in that PC space. Here PC1 reflects whether the gene express or not, PC2 reflects the heterogeneity of the two datasets, PC3 reflects the heterogeneity of different neurons. From plot A and plot B we can qualitatively say that cells that have a large loading in PC1, PC2 and PC3, for example, GABAergic cells in week 9 (week_9.hNbGaba) will have a higher contribution to disease risk prior.

4) The presented improvements in error rate control/statistical power, how do they translate to real life scenarios? I think it would help the reader to conceptualise what for example the improvement in recall in VBASS in Figure 3B really means.

A proper error rate control, as shown in original Figure 3A, means that the estimated error rate from our model is close to the real error rate. This is a basic assessment of the rigor of a statistical method. We added another sentence in section “VBASS showed better power than extTADA on simulated data with scRNA-seq expression” to further explain that, “Which means that the model estimated error rate from our model is close to the real error rate. In other words, if our model identified 100 genes at significance level 0.05 as estimated by our model, the real error rate will be close to 5%, which is 5 false positives.”

In this context, a higher statistical power means under same false discovery rate, VBASS could identify more disease risk genes. We added another sentence in the same section to explain that, “In other words, under same significance level (0.05 or 0.1), our model can identify more genes, while the error rate remains the same, as stated in (Figure 3A).” With those two features together, our model can identify more true disease risk genes at same sample size.

5) The authors assume that all disease genes belong to the same effect size distribution parametrized by 2 shared hyperparameters. Is this assumption valid based on known disease genes?

We acknowledge this is a simplified assumption that is effective but not necessary entirely valid. There could be qualitative difference in effect size across risk genes. However, we note that it is still a prior distribution. The variance of the distribution allows gene-specific, different values of posterior effect size implicitly or explicitly calculated based on data. Therefore, it is an effective strategy that has been used in previously published methods such as TADA and extTADA and related methods. We mentioned it again in the Method section “The probabilistic mode of VBASS” paragraph 4 as, “We assume \bar{k}_v and $\bar{\theta}_v$ are shared across all disease risk genes to reduce the number of parameters, similar to the assumption in TADA and extTADA”.

6) In the KL penalty term how is the pi-hyperparameter determined for the average proportion of disease risk genes? Does the choice of this parameter influence the results significantly?

The choice of this KL penalty term reflects the overall average proportion of disease risk genes that we expect in whole genome. This is a standard component in variational inference methods used for estimating parameters. This term will influence the result, where a larger value can lead to an underestimated FDR. A smaller value can lead to an overestimated FDR, as shown in the figure below (Supplementary Figure 5). We added that in the method section “Parameter inference for VBASS”:

we recommend two settings to avoid converging issues and keep good control of false discovery rate. First is setting the hyperparameter of KL penalty to the observed average proportion of risk genes, in practice we estimate that using extTADA. We showed that either too high or too lower KL penalty will result in improper false discovery control (Supplementary Figure 5). In practice, we suggest setting the KL penalty as estimated π value from extTADA or

the estimated genes that contribute to disease from any validated methods.

Supplementary Figure 5. The impact of KL penalty on false discovery control in simulation data. Each line represents the estimated false discovery rate (FDR, x-axis) and the real FDR (y-axis). We recommend setting the hyperparameter of KL penalty to the average of real disease risk prior, in other words, average proportion of disease risk genes. Either too high (blue, purple) or too low (yellow, red) kl penalty will result in improper false discovery control.

Minor:

1) Given that the authors compare their tool to extTADA, it would be useful to include a section of the introduction which describes the differences/similarities between the two methods

The major difference between two methods is that extTADA does not use gene expression data as input. This point has been discussed in details within following sections so we added a sentence in the introduction to briefly mention that as, “We compared the performances of VBASS with extTADA, a state-of-the-art Bayesian method which does not use expression data as input, under two conditions (bulk and scRNA-seq data) by both simulated and published de novo variants datasets to assess error control and statistical power.”

2) Currently the authors collapse the single-cell data per cell type into a vector which is then used as input into the NN to predict pi, instead of using the complete cell-wise count matrix which might allow the model to take into account within-cluster transcriptional heterogeneity. Wouldn't this additional information be useful for the prediction?

This is indeed an interesting idea. Capturing heterogeneity within each cell type or rare transient cell states could be very informative to disease risk. However, using count matrix in the model would add many more parameters (millions). Given the limited number of human

genes, training the model with millions of parameters will be extremely challenging and prone to overfit. We do agree there is room for improvement in how to represent and reduce the dimension of single cell data for genetic analysis.

3) It's not clear to me what the authors mean by the last few sentences of page 6 when they discuss this p-value based approach.

As the main scope of the paper is to use gene expression to estimate the prior for statistical tests and improve power in a Bayesian framework, we mention those frequentist (p-value based) approach to indicate that there are other ways to improve power, but those approaches have certain limitations.

Reviewer #2:

Summary:

This paper presented a Bayesian mixture model called VBASS with priors trained from single cell gene expression by the neural network to model the number of genetic variants. The author indicated that their method can be applied to both single cell data and bulk level data by replacing the neural network with a sigmoid function. The paper compared VBASS with a published state-of-art computational method extTADA on two types of datasets, which are simulated datasets and published real datasets. The author showed the model's statistical power by increasing sample size/gene size and showed its ability to identify novel risk genes by comparing candidate risk genes found by VBASS and extTADA and validating based on literature search.

Major comments:

1. Something I found unclear in the paper is how exactly single-cell data as opposed to bulk is helping the model. It is clear that the expression profile of a given gene across the cell types is used to determine the prior probability of being a risk gene, but the authors don't elaborate on how this benefits the model. For example, would a gene with a high level of expression across all cell types be considered more important than one that is highly variable or lowly expressed? I think clarifying what kind of profiles are expected to give a higher prior (or at least what profiles they found gave a higher prior) would help here.

No, we don't assume a gene with high level of expression across all cell types to be risk gene. The dependence of prior on expression profile is learned based on the data. We added a new Supplementary Figure 3 with the heatmap plot of gene expression of disease risk genes and non-risk genes, a gene will have a high risk prior only if the gene has high expression in some specific cell types while not others (like PAX5 and EBF3). A gene with high expression in all cell types will result in a moderate risk prior (like CSDE1, PBX1, etc.). A gene with low expression in all cell types will result in a low risk prior (like CCDC114, IGSF10, etc.). We mentioned that point in the result section "VBASS model disease risk association with both genetics and expression data" as, "Generally, VBASS will assign a higher disease risk to genes with relatively high expression in disease associated cell types while low expression in non-associated cell types, and vice versa."

2. The authors mention that the method only finds the correlation between cell types and risk genes and does not infer causality. It would be helpful if the authors discussed what changes they would have to make to enable to model to infer that. This is particularly important as being able to identify the relevant cell types would help justify the use of single cell data. For example, could they use a cell type-specific prior?

We thank the reviewer for suggesting this point. There are two issues with inferring causality. First, while one can test whether a single cell data set is informative about gene risk, it is very likely that the data set does not capture all important cell types and states, especially in terms of time points and anatomical regions (for complex organs). The information is often from cell types that are correlated with actual causal cell types in expression pattern. Second, even with better coverage of cell types and states from emerging cell atlas projects, we can only propose candidate causal cell types. Establishing causality would require functional studies such as conditional knockout in model organisms or human organoids.

3. It would have been interesting if the authors had compared the bulk VBASS method to the single-cell approach. This could have helped show if /how the use of single-cell data allows for the identification of more risk genes and/or narrows down what cell those genes are relevant in (i.e run analysis with pancreas bulk vs. single-cell pancreas).

The bulk VBASS version will be useful for conditions where bulk gene expression of corresponding organ is informative, such as the mouse expression data for congenital heart disease (as shown in previous publications such as Homsy et al 2015). When high-quality single cell data is available, the single cell version allows us to leverage higher resolution of expression profile to further improve power. To compare the two versions, we generated pseudo bulk RNA expression data via aggregation of the single cell dataset we used in autism and ran the bulk VBASS version to compare. Some well-known genes are not significant anymore, including *TSC2*, *CSNK2B*, *SPRY2*, *ZMYND8*, because they are specifically expressed in a small number of cell types (Supplementary Figure 3) and bulk dataset failed to show such specificity. We included that result in the section “VBASS identified novel ASD candidate risk genes on published DNV data”, as “To further validate that single cell data is more informative than bulk data in VBASS, we compared the results with bulk version VBASS predictions using pseudo bulk expression data that curated from the single cell data (Methods). As expected, in complex conditions like autism, bulk version VBASS failed to detect some well-known risk genes like *TSC2*, *CSNK2B*, *SPRY2* and *ZMYND8* (Table S6), because it does not have the information to model the correlation between cell type heterogeneity and disease risk.”, as well as in the discussion part paragraph 3 as “Generally, single cell version is more informative than bulk version, especially for complex conditions like autism, where bulk version of VBASS failed to detect the correlation between cell type heterogeneity and disease risk thus failed to detect some well-known risk genes.”. As well as in Supplementary Table S6.

4. The authors compared their method to extTADA which seems to be the gold standard for integrating scRNA-seq with DNV data for risk gene discovery. Only having one method used for comparison limits our ability to grasp how well the method performs as there are few reference points. Assuming that this was due to the lack of single cell approaches, the authors could see if there are bulk RNA methods, they could compare to bulk VBASS to have more benchmarking comparisons. Given the similarity between the bulk and single-cell VBASS, this could help prove the reliability of their model.

extTADA is a Bayesian method that only use DNV data for risk gene discovery while does not integrate scRNA-seq. To our knowledge, there is no method that use single cell or bulk RNA data jointly with de novo mutation data for risk gene discovery to benchmark with.

5. In the section "...identified novel CHD candidate risk genes on published DNV data", the author compared VBASS with the original TADA method. What is the intuition here to compare with the original TADA instead of extTADA?

This was in fact a typo. The method we compared with was extTADA. We have changed the sentence accordingly. TADA and extTADA have the same statistical models and use the same type of data (de novo mutations); their difference is the way they estimate the parameters. TADA estimate the parameters in a heuristic way, while extTADA used MCMC.

6. They have compared the results of their method with only one another approach. It would be good to compare with a few more approaches, if possible, such as mTADA, EncoreDNM.

Thanks for the reviewer's advice. We do note that the methods mentioned by the reviewers, including mTADA, EncoreDNM, all aim to increase statistical power in identifying risk genes, but we argue these methods are not really comparable to ours as they are based on completely different sources of information. mTADA leverages pleiotropic effects of risk genes in multiple conditions. It takes de novo mutation data from studies of different but related disorders. It does not consider gene expression profile. EncoreDNM aims to quantify genetic correlation among multiple conditions based on de novo mutation data. It does not identify individual risk genes or consider gene expression profiles. Therefore, we do not benchmark against them.

In the revision, we mention them in the background section, paragraph 1: "Several recently published methods attempt to increase power using additional information. EncoreDNM, m-TADA and M-DATA are statistical models that improve power by leveraging pleiotropic effect of risk genes across. DECO integrates pathways and gene sets information to prioritize risk genes. However, none of them tried to integrate the fetal single cell gene expression data information into rare variants analysis. Here we describe a method that integrates gene expression data of normal tissues with genetic data to improve power of finding new risk genes through rare variants."

7. The author showed that they can identify novel ASD risk genes on published data by removing known risk genes before training the model. How is VBASS's performance in identifying known risk genes? It would be more comprehensive if the author could add a section stating the model's ability to identify known risk genes on the published dataset.

We didn't remove known risk genes before training the model and calculate posterior probability of association. We only remove them when we estimate false discovery rate (FDR). FDR of a gene is the overall false discovery rate among genes with evidence at least as strong as the gene of interest. Most of known risk genes, due to their large effect sizes, have large PPA, as we shown in the new Supplementary Figure 6 of the PPA distribution in VBASS and extTADA. If we keep the known risk genes in the data, the FDR estimates for novel risk genes will be smaller than true values, which lead to over-confidence of novel discoveries. We added this part in method section "Parameter inference for VBASS":

For the real dataset, we removed the known risk genes selected as positives in training, which should have very large PPA due to their large effect sizes (Supplementary Figure 6), when we estimate FDR to identify candidate novel risk genes.

Supplementary Figure 6. Distribution of PPA value in VBASS and extTADA. Genes used in semi-supervised VBASS training were marked as blue and green, for positive and negative respectively. Other genes were marked as red.

8. Since the method can be applied on both single cell RNAseq and bulk RNAseq datasets, it would be nice to compare the results from a case where both scRNAseq and bulk RNAseq data were drawn from the same sample. This would provide an idea of how important it is / or not to have access to the high-resolution, high-cost scRNAseq datasets for such analyses.

Thanks for the suggestion. We've included the analysis in comparing the two version of VBASS. We note it is challenging to find such dataset with both scRNAseq and bulk RNAseq on the same sample relevant to the genetic data used in this study, thus we generated pseudo-bulk RNA expression data through aggregation of the single cell dataset of the brain tissues and ran the bulk VBASS version to compare in section "VBASS identified novel ASD candidate risk genes on published DNV data". In the bulk version of VBASS, some well-known genes are not significant anymore, including *TSC2*, *CSNK2B*, *SPRY2* and *ZMYND8*, because they are specifically expressed in a small number of cell types (Supplementary Figure 3) and bulk dataset failed to show such specificity. We included that result in the section "VBASS identified novel ASD candidate risk genes on published DNV data", as "To further investigate whether single cell data is more informative than bulk data in VBASS, we compared the results with bulk version VBASS predictions using pseudo bulk expression data that curated from the single cell data (Methods). As expected, in complex conditions like autism, bulk version VBASS failed to detect some well-known risk genes like *TSC2*, *CSNK2B*, *SPRY2* and *ZMYND8* (Table S6), because it does not have the information to model the correlation between cell type heterogeneity and disease risk.", as well as in the discussion part paragraph 3 as "Generally, single cell version is more informative than bulk version, especially for complex conditions like autism, where bulk version of VBASS failed to detect the correlation between cell type heterogeneity and disease risk thus failed to detect some well-known risk genes.". As well as in Supplementary Table 6.

Minor comments:

1. Grammar and Spelling: There were a few errors across the manuscript. I did not cite them here as I am not sure it is necessary for this stage of the review but can send them if required.

We've reviewed and revised all the grammar and spelling issues across the manuscript.

2. The introduction should discuss the different types of gene variants and their impact, to provide a better context of the biological research area

We expanded the introduction that there are different types of variants with largest effect sizes to disease and the difficulties of the small statistical power of using them in the first paragraph, "This is because that de novo coding variants with large effect sizes, including gene disruption variants and damaging missense variants, usually have low mutation rates and very low allele frequency in a study".

3. The introduction should outline and discuss the other methods developed in this area, to provide a better context of the existing methods.

Thanks for the reviewer's advice. We outlined other methods in this field in the first paragraph, "Several recently published methods attempt to increase power using additional information. EncoreDNM, m-TADA and M-DATA are statistical models that improve power by leveraging pleiotropic effect of risk genes across. DECO integrates pathways and gene sets information to prioritize risk genes. However, none of them tried to integrate the fetal single cell gene expression data information into rare variants analysis. Here we describe a method that integrates gene expression data of normal tissues with genetic data to improve power of finding new risk genes through rare variants."

Reviewer #3:

Zhong and co-author introduce VBASS, a Gamma-Poisson model to predict gene disease association from GWAS data. A novel aspect of the work is the use of gene expression data (from standard or single cell sequencing) as a prior for the Gamma-Poisson model. This is overall a sound and relevant method. They use simulated data to evaluate the statistical power of the approach and to compare it with a single competing method (extTADA). They also present the application of the method in two distinct real data sets (on coronary heart disease and Autism spectrum disorder), where they show that genes only detected by their method are of potential functional relevance. This is an overall interesting study, it fails short in a few important details.

1. The proposed model is quite intricate and explained with typical graph model representation in figure 1. Given that the Comms. Biology” has a more general readership, authors should make an effort to make figure 1 more conceptual, i.e. explanation of the parameters / input data / random variables.

We thank the reviewer for this suggestion. We would like to keep the Figure 1 as a supplementary figure because it demonstrated the graphic model where nodes represent parameters/variables and arrows indicate dependency. We revised the figure description to make it easier to understand. We made a new Figure 1 to show the model input, parameters, and model output, with numerical examples to help explaining the concepts to general readership.

2. There are not much details on how single cell gene expression is used. For the DNV study, I could only find a simple passage “single cell RNA-seq data of human fetal midbrain and prefrontal cortex from two publications.”. How are the expression vectors x_g parametrized? All individual cells independently of the cell type? How biased would the model be towards more abundant cell types? How is the data processed? Authors should expand this explanation and have a materials section dedicated to these. The same applies for the CHD analysis.

We thank the reviewer for pointing this out. In the revision, we expanded the description of data input in the method section: “*De novo* variants (DNV) and gene expression data”: “For CHD analysis using bulk version of VBASS, we input the gene expression rank value, ranged from 0-1, for each gene.”, and “For ASD analysis using single cell version of VBASS, we obtained single cell RNA-seq data of human fetal midbrain and prefrontal cortex from two publications. We used the combination of developmental time and cell ontology annotations as described in the two publications to define cell types. Small clusters of cell types with less than 10 single cells were removed. For each gene, we calculate the proportion of cells that express the gene in each timepoint specific cell types, ranged from 0-1, as the input vector to VBASS.”

3. The manuscript and benchmarking only mentioned one alternative competing method. There are several other similar methods in the literature (see below for a few examples), which are not cited. Authors need to put all competing methods in perspective and evaluate relevant methods in the benchmarking analysis.

<https://www.ncbi.nlm.nih.gov/pmc/articles/PMC5738153/>

This is extTADA and we have cited in ref.25 and benchmarked with it throughout our manuscript.

<https://journals.plos.org/plosgenetics/article?id=10.1371/journal.pgen.1009849>

This is M-DATA.

<https://academic.oup.com/bib/article/22/5/bbab067/6206366>

This is DECO.

We thank the reviewers for the suggestions for benchmarking analysis. The three methods mentioned are extTADA, M-DATA and DECO. We already included extTADA in the benchmarking analysis, while for the other two, we note that they were both trying to increase statistical power from different aspects. M-DATA focused on analyzing DNM data on multiple

traits and DECO focused on using gene-set enrichment analysis to prioritize disease risk genes. None of them focused on using gene expression data to increase statistical power. We decide not to benchmark against them but mention them in the background section, paragraph 1, “Several recently published methods attempt to increase power using additional information. EncoreDNM, m-TADA and M-DATA are statistical models that improve power by leveraging pleiotropic effect of risk genes across. DECO integrates pathways and gene sets information to prioritize risk genes. However, none of them tried to integrate the fetal single cell gene expression data information into rare variants analysis. Here we describe a method that integrates gene expression data of normal tissues with genetic data to improve power of finding new risk genes through rare variants.”

4. In the same line, authors should evaluate a variant of the VBASS, where no gene prior information is provided.

We thank the review for suggesting this variant of VBASS, but we note that this version should be very similar to extTADA, as they share the Poisson likelihood models of de novo mutation counts.

Reviewers' comments:

Reviewer #1 (Remarks to the Author):

Overall, most of my remarks have been adequately answered. I do have some remaining concerns:

- On the question of overfitting and generalization, are the genes used in semi-supervised pretraining included in the presented precision/recall estimates? The higher risk estimates shown in Figure 3C could indeed be because they a priori select genes with the highest risk priors, but this could also be inflated because the network is effectively overfitting on these genes and therefore if these genes are included in the precision/recall estimates this would lead to an overestimation of performance. I would have preferred a more in-depth study of model generalization, including proper cross-validation setups (e.g. leave out some genes which have been simulated to have high risk priors) as to assure no information leakage between training and testing, instead of the qualitative PCA-based argumentation the authors now included.

- On the new Supplementary Figure 3, it's not clear from this figure if there are particular cell types which are more or less influencing the disease risk priors. It would be clearer if the rows would be ordered according to $\log(\pi)$ value.

- Fig2: Figure legends should state that the subplot titles are different sample sizes. Also Fig2A wasn't showing any data on the page with the figure legend, but it did show the data properly on page 23 of the merged PDF.

- The estimation of the KL penalty hyperparameter requires running extTADA (line 360), why not include this calculation in the current model? It seems a bit superfluous that the user should have to run several tools to come to properly calibrated FDRs.

- On the new Supplementary Figure 3, it's not clear from this figure if there are particular cell types which are more or less influencing the disease risk priors (or even if there is a general difference in gene expression pattern distributions between genes under the null hypothesis vs risk genes). It might be clearer if the rows would be ordered according to $\log(\pi)$ value.

- A returning concern from the other reviewers is the limited comparisons to extTADA and not the other available tools. Although I understand the reasoning as to why not compare it to mTADA and other tools for more complex disorders (e.g. pleiotropy), if DECO includes prior information on pathways/genesets to assist prioritization it would be useful to compare their tool to it, even if it doesn't include gene expression information.

- The manuscript still contains many grammatical errors. But I'm confident the manuscript will go through a round of editorial proofreading before publication.

Reviewer #2 (Remarks to the Author):

My comments have been addressed.

Reviewer #3 (Remarks to the Author):

The authors have addressed all my raised points and I am satisfied with the current manuscript.

Response to reviewers' comments (comments in black, response in blue):

Reviewer #1 (Remarks to the Author):

Overall, most of my remarks have been adequately answered. I do have some remaining concerns:

- On the question of overfitting and generalization, are the genes used in semi-supervised pretraining included in the presented precision/recall estimates? The higher risk estimates shown in Figure 3C could indeed be because they a priori select genes with the highest risk priors, but this could also be inflated because the network is effectively overfitting on these genes and therefore if these genes are included in the precision/recall estimates this would lead to an overestimation of performance. I would have preferred a more in-depth study of model generalization, including proper cross-validation setups (e.g., leave out some genes which have been simulated to have high risk priors) as to assure no information leakage between training and testing, instead of the qualitative PCA-based argumentation the authors now included.

Yes, all genes are included in the precision/recall estimates. We removed the genes that were used during training to replot the precision-recall curve in the revised figure 3B. The results were similar, and conclusions remained the same. The reason is that we only included a small fraction of genes in training (100 positive and 300 negative) and the labels were only used during warm-up steps.

We revised the last sentence of methods section "Generation of simulation datasets":
"For VBASS, we randomly selected ~100 simulated risk genes and ~300 non-risk genes as label for the semi-supervised training. Those genes were excluded for comparison."

- Fig2: Figure legends should state that the subplot titles are different sample sizes. Also, Fig2A wasn't showing any data on the page with the figure legend, but it did show the data properly on page 23 of the merged PDF.

We updated the Figure2 legends with a sentence "Figure titles are the cohort sizes during simulation."

- The estimation of the KL penalty hyperparameter requires running extTADA (line 360), why not include this calculation in the current model? It seems a bit superfluous that the user should have to run several tools to come to properly calibrated FDRs.

This is a great point. The purpose of estimation of KL penalty hyperparameter is to find the average prior of all genes. This prior should be equal to the estimation when no gene expression data was given, which is exactly the assumption of extTADA. This is the reason we used it in the manuscript. VBASS can indeed estimate this hyperparameter when the input expression matrix is set to zero. We showed in Supplementary Figure 5 (now Supplementary Figure 6) that those two estimation methods have similar performances.

We updated the third sentence of method section "Parameter inference for VBASS" to:
"For the neural network version, we recommend two settings to avoid converging issues and

keep good control of false discovery rate. First is setting the hyperparameter of KL penalty to the observed average proportion of risk genes, in practice we recommend estimating that using extTADA or running VBASS without gene expression data. We showed that either higher or lower KL penalty will result in improper false discovery control (Supplementary Figure 5).”

We updated the figure legend of Supplementary Figure 5 (now Supplementary Figure 6) to: “The impact of KL penalty on false discovery control in simulation data. Each line represents the estimated false discovery rate (FDR, x-axis) and the real FDR (y-axis). We recommend setting the hyperparameter of KL penalty to the average of real disease risk prior, in other words, average proportion of disease risk genes, which can be inferred either from dry run of VBASS without expression data (pink) or from extTADA (green). Higher (cyan, blue) KL / real. π ratio will result in improper false discovery control, lower (yellow, red) KL / real. π ratio will result in stricter false discovery control and thus lower power.”

- On the new Supplementary Figure 3, it’s not clear from this figure if there are particular cell types which are more or less influencing the disease risk priors (or even if there is a general difference in gene expression pattern distributions between genes under the null hypothesis vs risk genes). It might be clearer if the rows would be ordered according to $\log(\pi)$ value.

We revised Supplementary Figure 3 (now Supplementary Figure 4) by breaking it into two plots. The first plot was ordered by $\log(\pi)$ and second was ordered based on both $\log(\pi)$ value and the hierarchical clustering result. We would like to note that order by $\log(\pi)$ is conflict with hierarchical clustering because π is a non-linear function of the expression values.

- A returning concern from the other reviewers is the limited comparisons to extTADA and not the other available tools. Although I understand the reasoning as to why not compare it to mTADA and other tools for more complex disorders (e.g., pleiotropy), if DECO includes prior information on pathways/gene sets to assist prioritization it would be useful to compare their tool to it, even if it doesn't include gene expression information.

We note that DECO use pre-defined gene sets (that are plausibly enriched with risk genes) to improve statistical power, whereas VBASS does not use pre-defined gene sets. Instead, it estimates the risk plausibility by integrating genetic data and gene expression data in a single graphical model. The other two reviewers agreed with the response to the comparison issues in the first revision. Nevertheless, we have now compared VBASS with DECO using de novo variant data from CHD and ASD. For DECO, we followed default protocols as described in the original publication (Nguyen et al 2021). Both DECO and VBASS identified more candidate risk genes than extTADA, but the extra risk genes identified by DECO are different to those identified by VBASS, and most of these extra risk genes are from a single gene set (GO term “Positive regulation of gene expression”). This result is expected, as these two methods use different sources of extra information to improve power. In general, DECO increases the power of discovering risk genes in pre-defined gene sets that are enriched with risk genes. VBASS use raw expression data, therefore the power improvement does not depend on prior knowledge of genes or gene sets.

We updated the last sentence of introduction:

“We also compared with DECO on published de novo variants datasets to assess the difference between gene expression data and curated gene set data for power improvement.”

The second last sentences in section VBASS identified novel CHD candidate risk genes on published DNV data:

“We noticed that DECO identified several other candidate risk genes when compared to VBASS (Supplementary Figure 3), but they showed few additional genetic evidence in the UK cohort (Table S5).”

The last sentence in paragraph 1 of section VBASS identified novel ASD candidate risk genes on published DNV data:

“Compared with VBASS, DECO identified a similar number of additional candidate genes. Most of these genes (25 out of 35) are from a single gene set (gene ontology term “Positive regulation of gene expression”). (Supplementary Figure 3, Table S8)”.

We added a paragraph in the Methods section:

“Comparison with DECO on CHD and ASD

We performed DECO on real CHD and ASD de novo variants dataset following the recommended protocols. We downloaded 21604 gene sets (≥ 5 genes) from MSigDB dataset (Version March 2020). We ran DECO for each gene set and selected the gene set with lowest p-value for comparison with VBASS. The gene sets were GO_HEART_DEVELOPMENT and GO_POSITIVE_REGULATION_OF_GENE_EXPRESSION for CHD and ASD, respectively. We calculated the Bayesian FDR using PPA for DECO in the same way as VBASS.”

- The manuscript still contains many grammatical errors. But I'm confident the manuscript will go through a round of editorial proofreading before publication.

We appreciate the comment. We have attempted to correct grammatical errors in several places.

Reviewer #2 (Remarks to the Author):

My comments have been addressed.

Reviewer #3 (Remarks to the Author):

The authors have addressed all my raised points and I am satisfied with the current manuscript.

REVIEWERS' COMMENTS:

Reviewer #1 (Remarks to the Author):

The authors have adequately adressed my remaining concerns.